# An Efficient Approach to the Five-Axis Flank Milling of Non-Ferrous Spiral Bevel Gears

**DOI:** 10.3390/ma14174848

**Published:** 2021-08-26

**Authors:** Hao Xu, Yuansheng Zhou, Yuhui He, Jinyuan Tang

**Affiliations:** 1Changsha Zhongchuan Gear & Transmission Driveline Co., Ltd., Changsha 410200, China; xuhao518@sina.cn; 2State Key Laboratory of High Performance Complex Manufacturing, Central South University, Changsha 410083, China; zyszby@csu.edu.cn (Y.Z.); jytangcsu_312@163.com (J.T.); 3College of Mechanical and Electrical Engineering, Central South University, Changsha 410083, China

**Keywords:** five-axis flank milling, spiral bevel gears, non-ferrous, CNC machining, tool path planning

## Abstract

Five-axis flank milling has been applied in industry as a relatively new method to cut spiral bevel gears (SBGs) for its flexibility, especially for the applications of small batches and repairs. However, it still has critical inferior aspects compared to the traditional manufacturing ways of SBGs: the efficiency is low, and the machining accuracy may not ensure the qualified meshing performances. To improve the efficiency, especially for cutting non-ferrous metals, this work proposes an approach to simultaneously cut the tooth surface and tooth bottom by a filleted cutter with only one pass. Meanwhile, the machining accuracy of the contact area is considered beforehand for the tool path optimization to ensure the meshing performances, which is further confirmed by FEM (finite element method). For the convenience of the FEM, the tooth surface points are calculated with an even distribution, and the calculation process is efficiently implemented with a closed-form solution. Based on the proposed method, the number (or total length) of the tool path is reduced, and the contact area is qualified. Both the simulation and cutting experiment are implemented to validate the proposed method.

## 1. Introduction

Spiral bevel and hypoid gears are significant components of power transmission systems used in automobiles, helicopters, etc. Spiral bevel gears perform rotation about intersecting axes, while hypoid gears rotate about crossed axes. Both spiral bevel gears and hypoid gears can be manufactured in the same way. In the following article, we only mention SBGs for the convenience. SBGs are mainly cut by three conventional approaches, face milling [1,2], face hobbing [3,4], and hobbing [5]. In some cases, the tooth surfaces should be modified by changing the machine setting or cutter motion. Achtmann and Bär [6] applied modified helical motion and roll to produce optimally fitted bearing ellipse. Fan [7] used higher-order polynomials to represent the cradle increment angle of the machine setting than the conventional way and developed TCA (tooth contact analysis) programs in the Gleason commercial software CAGE. Simon [8] reduced the transmission errors by defining the cradle radial setting and the cutting ratio with fifth-order polynomial functions and optimizing them. All of these approaches are equipped with special gear manufacturing machine tools, which rely on corresponding manufacturers.

Unlike the conventional approaches, computer numerical control (CNC) milling has also been introduced as a new technology to cut SBGs on general CNC machine tools. Although CNC milling has a lower production rate than the conventional approaches, it takes advantage of cutting SBGs for small batches, prototypes and repairs (When the gear is worn, it can be repaired by CNC milling). In addition, since the rigidity of CNC milling machine tools is poorer than the conventional gear machine tools, the cutting efficiency is limited to the application of CNC milling to the gears with the hard material. This limitation is not a serious problem for the non-ferrous metals. Hence, it would be beneficial to find the efficient way for CNC milling of non-ferrous gears.

With the recent development of CNC milling technologies, CNC milling of SBGs becomes an advanced technology applied in some gear manufacturers. Many researchers have studied the manufacturing of SBGs with CNC milling. Tsiafis et al. [9] studied design and manufacturing of SBGs using CNC milling machines. Li et al. [10] proposed a novel integrated design and machining mode of SBGs based on universal CNC machine tools. In order to solve the problem that the tooth surface points cannot cover the whole tooth surface, Wang et al. [11] proposed a new adaptive geometric meshing theory with an advanced study of cutter geometry and meshing theory. Shih et al. proposed a method to manufacture face-milled SBGs and the cutter head on a five-axis CNC machine [12] and then carried out real cutting experiments [13]. Álvarez et al. [14] studied different machining strategies of five-axis CNC milling to improve the machining quality of SBGs. Based on the updated Kriging model, Deng et al. [15] proposed a method to improve the accuracy of tooth surface reconstruction. Gleason’s UpGear method [16] is intended for CoSIMT-type tools on Gleason-Heller 5-axis CNC machines.

End milling is a mode of CNC milling to remove the material around the midst of the cutter’s flat end. DMG-Mori’s gearMILL [17] offers SBG modules cutting with end mill and ball mill tools. The alternative mode, flank milling, removes the material along the cutter flank. Comparing end milling and flank milling, flank milling takes advantage of quality enhancement, manufacturing time and cost reduction [18,19], and it has been widely applied to manufacturing a ruled surface [20]. For some of the existing practical models of SBGs, their working parts are ruled surfaces, or some are close to ruled surfaces. Taking the example of the gears cut by Gleason’s face-milled methods, which include non-generated and generated methods: the working part of the non-generated gear is a conical cutter surface, which is a ruled surface; the working part of the generated gear is the envelope surface of the conical cutter surface, and it is close to a ruled surface [21]. Hence, it makes sense to apply the flank milling to cut SBGs.

Flank milling and the manufacturing of SBGs are two different disciplines. Exksting research on flank milling is mainly about the minimization of machining errors. The corresponding issues include envelope surface [22,23,24,25], cutter-workpiece engagements [26,27,28,29,30], geometrical deviations [24,31,32], tool path planning strategies [33,34,35,36,37,38,39], tool path optimization with: particle swarm optimization method [40], dynamic programming method [41], local method [42], global method [43], constraints [44], generic cutters [45], the consideration of tool path smoothness [46], the influences of tool axis [47] and cutter runout [48,49], cutter optimization of shape [50] and size [51], etc. In contrast, the manufacturing of SBGs values the working performances [21], such as contact path and transmission errors. By considering the difference, Zhou et al. [52,53] focused on the machining of tooth surface area, and an extra pass (tool path) was needed to machine the bottom of a tooth slot of SBGs, so the machining efficiency can be further improved. Meanwhile, they did not give the solution of TCA for the flank milling of SBGs, and it is very difficult, due to the complicated tooth surface geometry, but important since, it directly evaluates the meshing performances of the machined SBGs.

In this work, a filleted end mill cutter is used to simultaneously cut the tooth surface and tooth bottom with only one pass, and this method improves the machining efficiency by avoiding the extra pass to cut the tooth bottom, which is necessary for the methods in [52,53]. In addition, compared with the traditional flank milling method considering the whole tooth surface errors, the tool path planning strategy and optimization model proposed in this paper focus on the machining accuracy near the tooth surface contact area, which can ensure that the tooth surface has better meshing performances. In order to verify the effectiveness and authenticity of the proposed method, the TCA with FEM is applied with a novel tooth surface modeling approach to check the meshing performances of the machined SBGs, and actual machining experiments are carried out. The analysis results show that the contact area is qualified. In Section 2, the tooth surface and meshing of SBGs are introduced. The optimization model of tool path planning is established in Section 3. With the planned tool path, a new closed-form representation is proposed to efficiently generate tooth surface points with an even distribution for the convenience of building the FEM model. Subsequently, the TCA of the flank milling of SBGs is carried out based on the FEM, as described in Section 4. The cutting simulation and experiments are explained in Section 5.

## 2. Tooth Surface and Meshing of SBGs

The 3D model of an SBG with 33 teeth is shown in Figure 1. Each tooth slot has both a convex side and a concave side. The depth of the tooth slot gradually decreases from heel to toe. Here we assume that the tooth surface is a given designed surface and represented as g(h,ϕ). The details of the tooth surface model can be referred to as [2,53].

When a pair of SBGs are meshing to transfer power during the process of two tooth surfaces engaging with each other, they contact at different points when they are treated as rigid bodies. Practically, they contact at small ellipses around the contact points due to the deformation of the tooth surface. The contact paths are formed by connecting these points, and the contact area is generated by joining these contact ellipse. Once both tooth surfaces of a pair of gear drive are obtained, the contact path and area can be calculated by TCA [21]. For a gear drive, edge contact should be avoided since it decreases work performance and serves life. A good design of tooth surface should be capable of avoiding the edge contact while considering practical circumstances, including manufacturing errors, load, and errors of alignment. Subsequently, an ideal contact path is usually chosen as the middle of a pair of tooth surfaces of a gear drive [21], as shown in Figure 1.

Since the tooth surface is a complex 3D surface, it is difficult to define the ideal contact path directly. Alternatively, it can be defined according to the gear blank. As shown in Figure 1, the ideal contact path is coincident to the middle of a pair of tooth surface. A point $q^{\prime }$ on the middle of a pair of tooth surface can be mathematically represented according to the gear blank parameters. When $q^{\prime }$ is defined by the gear blank parameters, the contact points on the tooth surface can also calculated according to a mapping approach as shown in Figure 1. When point $q^{\prime }$ is rotated along with $z_{g}$, a circular curve is formed and intersected with the convex side of tooth surface as q. Subsequently, the mapping relationship can be used to calculate the contact point on the tooth surface [52]. We have a system of two equations in two unknowns [52]: (1)$$\left\{\begin{array}{c} Z\left(h^{*},\phi^{*}\right)=z_{q}\\ X^{2}\left(h^{*},\phi^{*}\right)+Y^{2}\left(h^{*},\phi^{*}\right)=r_{q}^{2}. \end{array}\right.$$
where $X(h^{*},\phi^{*})$, $Y(h^{*},\phi^{*})$, $Z(h^{*},\phi^{*})$ are the coordinates of the tooth surface in $S_{g}$. Once both unknowns $h^{*}$ and $\phi^{*}$ are calculated according to Equation (Equation 1), the contact point on the contact path is obtained by submitting into the tooth surface g(h,ϕ).

## 3. Tool Path Planning Strategy and Optimization Model for Flank Milling of SBGs

### 3.1. Tool Path Planning Strategy

As shown in Figure 2, a process is considered to generate a cutter envelope surface tangent to the tooth surface along the contact path. At each instant of the process, the cutter surface is tangent to the tooth surface at a point, which is represented on both surfaces as p and q, respectively. The trajectories of the tangent point on both surfaces are cutter contact (CC) line p(ϕ) and contact path q(ϕ), respectively. Once p contributes as a point on the cutter envelope surface, it means the cutter envelope surface is also tangent to the designed surface at p. The necessary conditions of the cutter envelope surface tangent with the tooth surface along q(ϕ) are summarized in terms of two aspects [53].

The cutter surface is tangent to the tooth surface along q(ϕ).For p(ϕ), it must satisfy
(2)$$\frac{d\rho (h)}{dh}\cdot \frac{dh(\phi )}{d\phi }=0.$$

The left side of Equation (Equation 2) is directly determined by the CC line on the cutter surface. For a cylinder, ρ(h) is a constant and h(ϕ) can vary.

With both models of tooth surface and contact path, the tool path planning for five-axis flank milling is implemented with two steps. First, the tool path strategy based on the necessary conditions is used to make the cutter envelope surface tangent to the designed surface along the contact path. Second, cutter orientations are optimized to obtain the minimal geometric deviations of the contact area.

In order to improve the efficiency, the fillet end mill cutter is used to process the tooth slot once to ensure the one-time machining of tooth surface and fillet part. As shown in Figure 3, a filleted end mill cutter is used only one pass to machine the convex side of the tooth surface. According to the necessary conditions, the cutter is planned to be tangent to the tooth surface along the contact path. For a point q on the contact path, a local coordinate system $S_{q}\left(q;n,t,d\right)$ is established. n is the unit normal vector of both cutter envelope surface and tooth surface. t is the unit tangent vector of the contact path at q. d is obtained as d=n×t. Furthermore, the tool path planning strategy is stated with two aspects: the cutter axis l is a unit vector in the tangent plane tqd; cutter tip point $o_{c}$ is determined according to the tangent between the cutter and root cone.

As shown in Figure 3, the tool axis l could be a varied unit vector in the tangent plane, and it can be defined with an angle μ. $o_{c}$ is the tip point. With a given l, $o_{c}$ can be determined with a solution to *h* calculated according to the condition that the cutter is tangent with root cone, which will be explained later. We have
(3)$$\begin{array}{cc} l\left(\phi \right) & =\cos\left(\mu \left(\phi \right)\right)\cdot t\left(\phi \right)+\sin\left(\mu \left(\phi \right)\right)\cdot d\left(\phi \right)\\ o_{c}\left(\phi \right) & =p_{h}\left(\phi \right)-h\left(\phi \right)\cdot l\left(\phi \right)\\ \; & =q\left(\phi \right)+R\cdot n\left(\phi \right)-h\left(\phi \right)\cdot l\left(\phi \right) \end{array}$$
where *R* is the radius of the cutter. It should be noted that because the fillet part is directly machined by the bottom of the cutter, it is necessary to ensure that the toroidal surface of the cutter is tangent to the root cone. As shown in Figure 4, the distance between $p_{h}$ and the tip point is $h_{m}$, the distance between $p_{h}$ and the root cone is ($h_{q}$+c)/2 and marked as *M*. Therefore, the following equation holds
(4)$$\left\{\begin{array}{c} h_{m}=M\sin\mu +r\left(1-\sqrt{1-{\left(\frac{M\cos\mu -d}{r}\right)}^{2}}\right)\\ M=\frac{h_{q}+c}{2} \end{array}\right.$$
where *r* is the fillet radius of cutter and *d* is the cutter constant. When the toroidal surface is tangent to the root cone, $h_{m}$ can be approximately regarded as *h* in Equation (Equation 3). Therefore, *h* in Equation (Equation 3) can be replaced by $h_{m}$ in Equation (Equation 4) to obtain the coordinates of tip point $o_{c}$.

Because n, t, and d can be calculated in coordinate system $S_{g}$, l and $o_{c}$ can also be obtained in $S_{g}$ by transforming from $S_{q}$. Since μ is the variable to define l, a polynomial function in terms of the motion parameter ϕ is applied to define the μ as
(5)$$\mu \left(\phi \right)=u_{1}+u_{2}\cdot \phi +u_{3}\cdot \phi^{2}+u_{4}\cdot \phi^{3}$$
where $u_{i}\left(i=1∼4\right)$ are the coefficients of the polynomials. Assuming $x=[u_{1},u_{2},u_{3},u_{4}]$, the optimization problem with respect to the variable x is proposed as follows to minimize the geometric deviations of the contact area.

In summary, the tool path planning strategy generally meets the following conditions.

The necessary condition of the cutter envelope surface tangent to the designed tooth surface is satisfied.The constraint condition of the toroidal surface of the cutter tangent to root cone is satisfied.The constraint condition that the swing angle of tool axis μ is a function of motion parameter ϕ is satisfied.

### 3.2. Optimization Model to Minimize the Geometric Deviations of the Contact Area

Once the cutter envelope surface and tooth surface are tangent along the contact path, both of them have the same normal curvature at every point on the tangent direction of the contact path. An effective way to reduce the geometric deviations around this point is to minimize the relative normal curvature of direction d [53]. Furthermore, for the geometric deviations of the contact area, it is an effective way to minimize the overall relative normal curvatures of direction d along the contact path. By inserting Equation (Equation 3) into Equations (Equation 10) and (Equation 11), the cutter envelop surface can be obtained. When the representations of theoretical tooth surface and cutter envelope surface are known, the normal curvature of any point on the surface along any direction can be calculated according to the theory of differential geometry. For a point **q**(*u*,*v*) on the contact path, the principal curvature and principal direction at this point can be calculated by Equation (Equation 6).
(6)$$\begin{array}{cc} \; & \mu_{1,2}=\frac{-\left(LG-NE\right)\pm \sqrt{{\left(LG-NE\right)}^{2}-4\left(LF-ME\right)\left(MG-NF\right)}}{2(LF-ME)}\\ \; & L\mu_{1,2}+M=k_{1,2}\left(E\mu_{1,2}+F\right)\\ \; & e_{1,2}=\frac{r_{u}\mu_{1,2}+r_{v}}{\left|r_{u}\mu_{1,2}+r_{v}\right|} \end{array}$$
where L, M, and N are the first fundamental homogeneous coefficients, and E, F, and G are the second fundamental homogeneous coefficients. $k_{1,2}$ and $e_{1,2}$ are the principal curvature and the principal direction at **q**, respectively. Suppose that the angle between the tangent direction **d** and the principal direction $e_{1}$ at point **q** is θ, as shown in Figure 5. Then, according to Euler formula, the normal curvature of point **q** along the tangent direction **d** is
(7)$$\begin{array}{c} k_{d}=k_{1}\cos^{2}\theta +k_{2}\sin^{2}\theta \end{array}$$

In conclusion, for a set of data points $q_{i}\left(i=1∼N\right)$ sampled from the contact path, the following model can be used to describe the error between the cutter envelope surface and the designed tooth surface.
(8)$$\begin{array}{cc} \; & \min\sum_{1}^{N}(k_{d}-k_{e}(x){)}^{2}\\ \; & s.t.\;\;\mu_{min}\leq \mu_{i}\leq \mu_{max},i=1∼N\\ \; & h_{m}=M\sin\mu +r\left(1-\sqrt{1-{\left(\frac{M\cos\mu -d}{r}\right)}^{2}}\right),M=\frac{h_{q}+c}{2}. \end{array}$$
where x is optimal variable; $\mu_{min}$ and $\mu_{max}$ are used to define the range of $\mu_{i}$; $k_{e}$ and $k_{d}$ are the normal curvatures along the direction d of designed surface and cutter envelope surface, respectively. In some cases, the absolute value of relative normal curvature is close to 0, so the minimum relative curvature radius can be used to describe the closeness between two surfaces, as shown in Figure 6.

The corresponding optimization model is replaced by
(9)$$\begin{array}{cc} \; & \min\sum_{1}^{N}{\left(\frac{1}{k_{d}}-\frac{1}{k_{e}\left(x\right)}\right)}^{2}\\ \; & s.t.\;\;\mu_{min}\leq \mu_{i}\leq \mu_{max},i=1∼N\\ \; & h_{m}=M\sin\mu +r\left(1-\sqrt{1-{\left(\frac{M\cos\mu -d}{r}\right)}^{2}}\right),M=\frac{h_{q}+c}{2}. \end{array}$$

## 4. Modeling and Contact Analysis of SBGs

In order to ensure the good contact performances of the tooth surface after flank milling, the contact analysis of the tooth surface based on the FEM can be carried out, so the three-dimensional model of the tooth surface is needed. In order to facilitate the modeling in an efficient way, tooth surface points should be generated with an approximately even distribution [54]. Although [54] gave an effective way to achieve this goal, a complicated computation algorithm was applied to solve a global optimization problem. In contrast, we here give a new way, which directly applies closed-form calculations to efficiently generate the tooth surface points.

Assume that the cutter surface is expressed as r(h,θ), where *h* and θ are two independent parameters. When the cutter surface moves continuously along a tool path defined with parameter ϕ, a family of surfaces of the cutter surface is generated. The envelope surface is the boundary of the family surfaces. The envelope surface can be calculated as a closed-form result according to the geometric meshing theory (or geometric envelope approach) [2,31] as
(10)$$r\left(h,\phi \right)=o_{c}\left(\phi \right)+h\left(\phi \right)\cdot l\left(\phi \right)+\rho \left(h\right)\cdot n\left(h,\phi \right)$$
where, as shown in Figure 2, the cutter surface is represented as a surface of revolution; $o_{c}$ is the cutter tip point; n is the unit normal of the cutter surface at a point p; ρ is the distance between p and $p_{h}$, which is the intersected point of n and l, the cutter axis; α is the angle formed by n and l. n in Equation (Equation 10) can be calculated as as [2,31]
(11)$$n\left(h,\phi \right)=\frac{\cos\alpha \cdot v_{h}^{2}}{{\left(l\times v_{h}\right)}^{2}}\cdot l-\frac{\left(l\cdot v_{h}\right)\cdot \cos\alpha }{{\left(l\times v_{h}\right)}^{2}}\cdot v_{h}\pm \frac{\sqrt{{\left(l\times v_{h}\right)}^{2}-\cos^{2}\alpha \cdot v_{h}^{2}}}{{\left(l\times v_{h}\right)}^{2}}\cdot \left(l\times v_{h}\right)$$
where $v_{h}$ is the velocity of $p_{h}$.

However, for some special cases, Equations (Equation 10) and (Equation 11) are not appropriate to obtain the contact points covering the whole tooth surface, which will be mentioned later. Now, another method of calculating the envelope surface is introduced by taking the filleted end mill cutter as an example.

As shown in Figure 7, assume that $p_{2}$ is a contact point on the toroidal surface, $n_{2}$ denotes the unit normal of the cutter surface at $p_{2}$, t is a unit vector on the plane determinated by $p_{2}$ and l, and t is orthogonal to the tool axis l. The geometric characteristic can be expressed as
(12)$$n_{2}\cdot \left(e\times l\right)=0$$

Combining with this geometric characteristic and envelope condition, we can obtain the contact points on the toroidal part, which will be described next.

As shown in Figure 7, $p_{2}$ can be expressed as
(13)$$p_{2}=o+rl+\left(R-r\right)e+rn_{2}$$
and the velocity $v_{p2}$ on point $p_{2}$ can be obtained as
(14)$$v_{p2}=v_{o}+w\times \left(p_{2}-O\right)=v_{o}+w\times \left(rl+\left(R-r\right)t+rn_{2}\right)$$

Considering the envelope theory, the envelope condition can be written as
(15)$$n_{2}\cdot v_{p2}=n_{2}\cdot \left(v_{O}+w\times \left(rl+\left(R-r\right)t\right)\right)=0$$

Assume that $p_{h}$ is the center point of the circle, which is the intersection of cylindrical part and toroidal part, the velocity of point $p_{h}$ is $v_{h}$. Hence, Equation (Equation 15) can be rewritten as
(16)$$n_{2}\cdot \left(v_{h}+\left(R-r\right)\left(w\times t\right)\right)=0$$

According to the Equations (Equation 12) and (Equation 16), $n_{2}$ can be obtained as
(17)$$n_{2}=\pm \frac{\left(l\times t\right)\times (v_{h}+\left(R-r\right)\left(w\times t\right))}{\left|\left(l\times t\right)\times (v_{h}+\left(R-r\right)\left(w\times t\right))\right|}$$

By substituting Equation (Equation 17) into Equation (Equation 13), corresponding contact points can be calculated. By sampling parameter t with all directions on the plane orthogonal to the tool axis l, the contact curve on the toroidal part can be obtained.

The envelope surface can be calculated by using the above two methods, but in some cases, one method may be more suitable than the other.

In Figure 8, the contact line on the cylindrical surface is mainly in the height direction. If the discrete *h* method is used, more contact points (blue points) will be obtained. If the discrete θ method is used, fewer contact points (green points) will be obtained. The contact line on the toroidal surface is mainly in the width direction. If the discrete *h* method is used, fewer contact points (blue points) will be obtained. If the discrete θ method is used, more contact points (green points) will be obtained. Therefore, in order to obtain contact points that cover the whole tooth surface uniformly, we need to select the appropriate method to calculate the envelope surface according to different situations.

According to the data in Table 1, combined with the tool path planning method and closed-form modeling method proposed in this paper, the optimization model is established according to Equation (Equation 9). For this model, the range of $\mu_{i}$ is chosen as $\left[\frac{\pi }{6},\frac{5\pi }{6}\right]$, and N=21. With the optimal result, the machined surface is calculated according to Equations (Equation 10) and (Equation 11) and modeled in CATIA V5R20, as shown in Figure 9. The 3D model was imported into ABAQUS to establish the FEM. The material properties are Young’s modulus 206,800 and Poisson’s ratio 0.3. Using the five tooth calculation model, both the gear and the pinion contain 131,600 elements, and the element type is solid element C3D8R. Checking the mesh quality indicates that both the error mesh and the warning mesh are 0. The torque is 2625 N·m, which is applied to the gear. Set the number of CPU cores to 30. After submitting the job, the analysis lasted for 6 h 20 min, and the analysis results are shown in Figure 10. There is a slight edge contact between the gear and the pinion, but the overall contact area is basically in the tooth direction.

## 5. Machining Simulation and Experiment

An SBG represented by the parameters in Table 1 was machined. The radius and fillet radius of the cutter are 2 mm and 1.5 mm, respectively. The cutter has been checked without interference on both sides of the working part of the tooth surface.

In the experiment, five slots were machined, the calculated tool path was imported into CATIA, and the reasonable advance and retreat tool path was set. Finally, the complete tool path of this experiment was obtained. The simulation processing was verified in Mastercam, as shown in Figure 11a,b. Finally, the five tooth slots model of SBG was obtained and compared with the theoretical tooth surface, as shown in Figure 11c.

The simulation results show that the tooth surface errors in the meshing area are between −0.01 and 0.01 mm, while the tooth surface errors in the top and root area are relatively large.

The corresponding NC code is calculated by using the simulation software, and the actual processing experiment is carried out on the five-axis NC precision engraving machine GR200V A15SH manufactured by Beijing Jingdiao Technology Co., Ltd. (Beijing, China). The gear material is 7075 aluminum alloy and the five tooth slots model of SBG is obtained, as shown in Figure 12.

The tooth profile errors of the machined tooth surface were measured by CMM (coordinate measuring machine). According to Gleason’s standard, 5 × 9 grid points are planned on the gear shaft section, and the tooth profile errors are calculated with the grid midpoint as the reference point, as shown in Figure 13. According to the tooth surface equation of SBG, the theoretical coordinates p and normal vector n of 5 × 9 grid points can be easily obtained. After the theoretical data are input into the CMM, the machine will automatically touch the tooth surface to obtain the actual measured coordinate values $p_{t}$, and the tooth profile errors can be calculated according to Equation (Equation 18).
(18)$$\delta =(p-p_{t})\cdot n$$

The tooth profile errors of concave and convex sides of tooth surfaces are shown in Figure 14a,b, respectively. The measurement results show that the errors of the middle part of the tooth height direction are relatively small, and the errors of the upper and lower sides gradually increase, which is consistent with the simulation results, and the maximum error is less than 20 μm. The errors of meshing area are larger than the simulation result, which is mainly due to the existence of certain machine tool errors and measurement errors. Referring to the five-axis machining center in Hermle, Germany, which is a world-class brand, the accuracy of machined parts is generally between 0.005 and 0.01 mm. The errors between the actual machined surface and the simulated surface are less than this order of magnitude. Therefore, the error fluctuation value is in a reasonable range, which shows the correctness of the given tool path.

## 6. Discussion

According to the tooth surface geometry and its meshing utilization, each side of the tooth surface of SBGs is cut by the five-axis flank milling with only one pass. A filleted end mill cutter is applied to implement the tool path planning. Due to the special requirements of gears in meshing performances, the contact area is accurately machined by the cutter flank side with a tangency condition between the cutter envelop surface and tooth surface along the contact path. The fillet part is directly machined by the cuter bottom. The tooth surface of flank milling can be obtained by two different explicit calculation methods. In some cases, one method may be more suitable than the other. By optimizing the tool path, the accuracy of tooth surface and contact performance can be improved. The machining efficiency is improved because each side of the tooth surface is machined by one pass rather than multiple passes.

In order to verify the effectiveness and authenticity of the method proposed in this paper, contact simulation analysis and flank milling experiments were carried out, respectively. The simulation results show that there is a slight edge contact between the gear and the pinion, and the contact trace is slightly closer to the inner diameter of the gear, but the overall contact area is basically towards the tooth direction. Theoretically, the contact trace should be located in the middle of the tooth surface. However, for the following two reasons, the results shown in Figure 10 are the best results that the proposed method can achieve at present.

The contact trace of SBGs are sensitive to the topological morphology of the tooth surface. Small changes in the tooth surface may cause changes in the position and orientation of the contact trace.The research content of this paper aims to adopt an efficient flank milling method to ensure low tooth surface error near the contact area, which is not a real research method to fully realize the pre-control of meshing performances.

Further research is needed to realize the pre-control of meshing performances of SBG based on five-axis flank milling. The experimental results show that the tooth surface error of the meshing area obtained by NC simulation is less than 10 μm, while the maximum error of the actual machining is less than 20 μm. According to the five-axis machining center in Hermle, Germany, which is a world-class brand, the accuracy of machined parts is generally between 0.005 and 0.01 mm. The errors between the actual machined surface and the simulated surface is less than this order of magnitude. Therefore, the error fluctuation value is in a reasonable range, which shows the correctness of the given tool path.

For some cases, the high machining accuracy is required for not only the contact area, but also the other areas. For those cases, multiple passes are needed to machine the convex or concave sides of the tooth surface, and some further machining processes might also be needed to accurately machine the fillet part of tooth surface. Furthermore, future work about practical machining, measurement, and error control would also be valuable.

## Figures and Tables

**Figure 1 materials-14-04848-f001:**
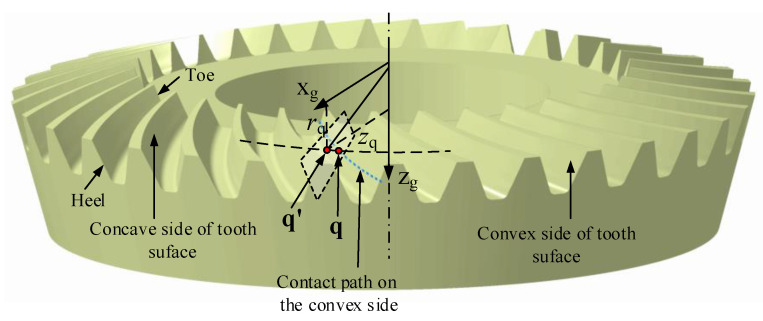
Tooth surface and contact path.

**Figure 2 materials-14-04848-f002:**
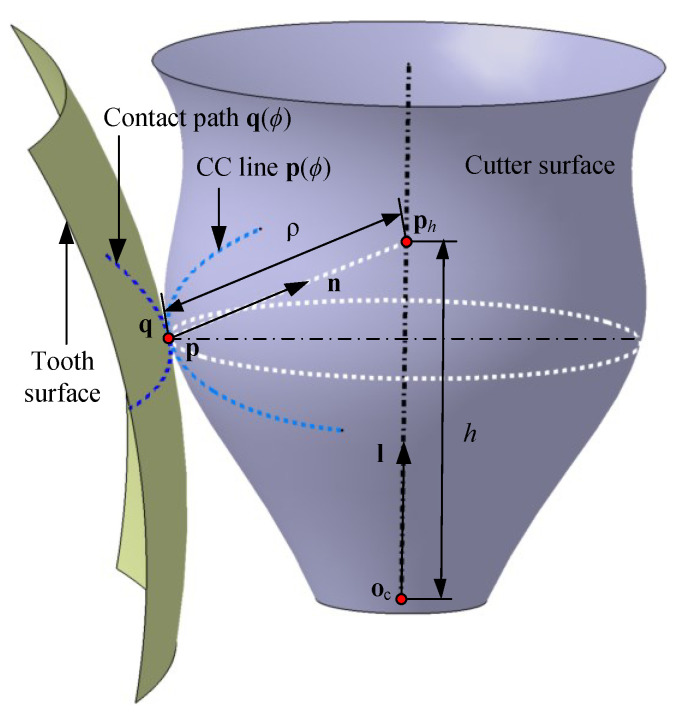
The tangency condition between the cutter envelope surface and tooth surface.

**Figure 3 materials-14-04848-f003:**
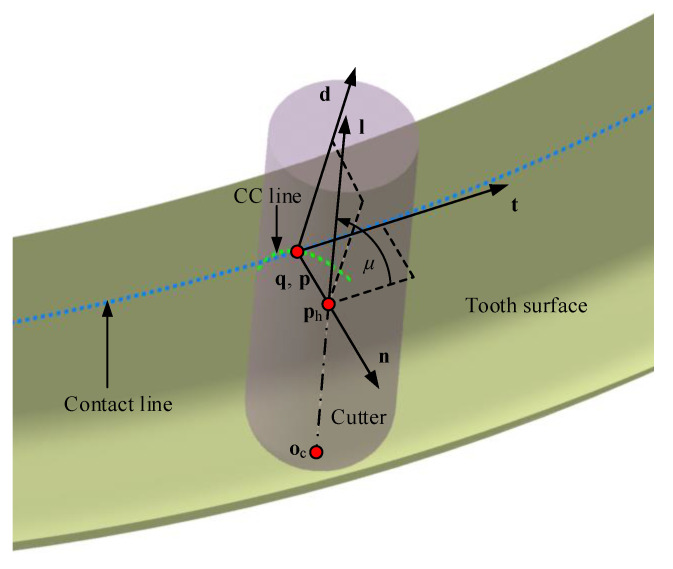
Tool path planning strategy.

**Figure 4 materials-14-04848-f004:**
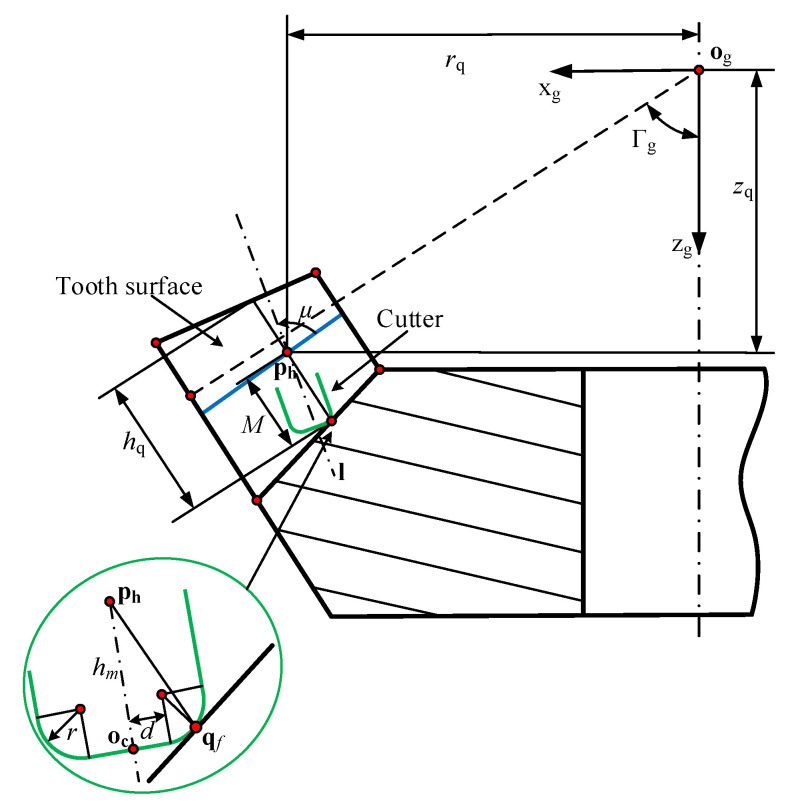
Cutter constraints in machining.

**Figure 5 materials-14-04848-f005:**
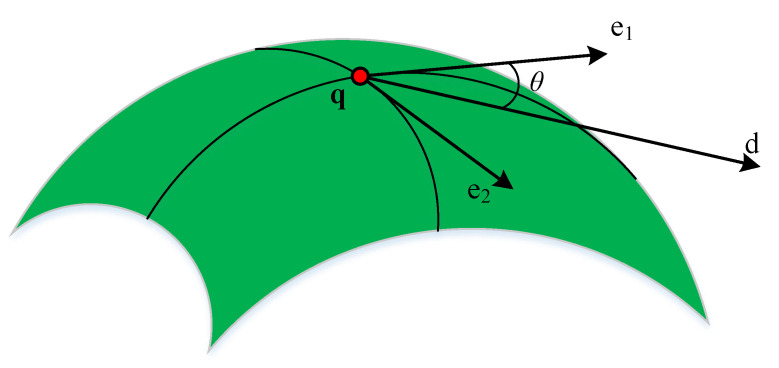
Principal direction and tangent direction.

**Figure 6 materials-14-04848-f006:**
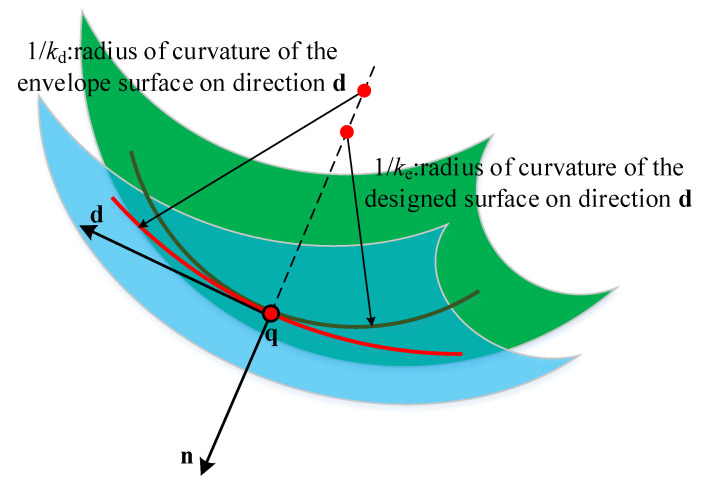
Description deviation of relative radius of curvature.

**Figure 7 materials-14-04848-f007:**
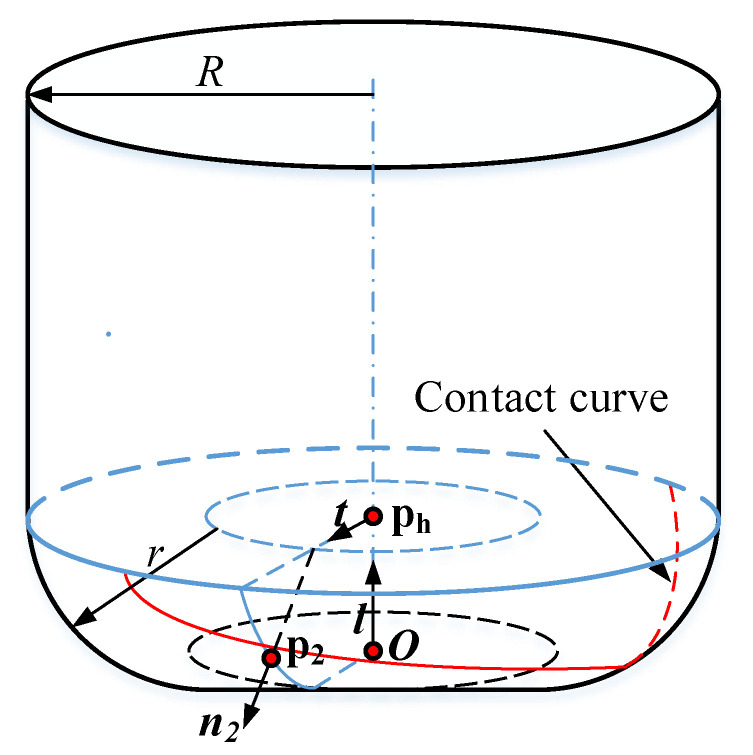
Calculating the contact point on the filleted end mill cutter.

**Figure 8 materials-14-04848-f008:**
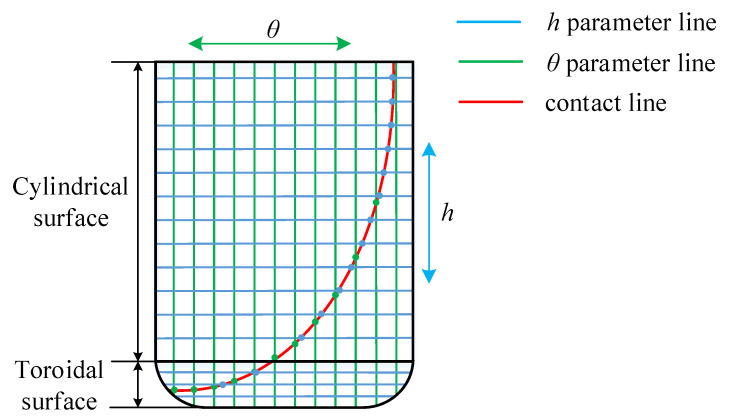
Calculation of contact line.

**Figure 9 materials-14-04848-f009:**
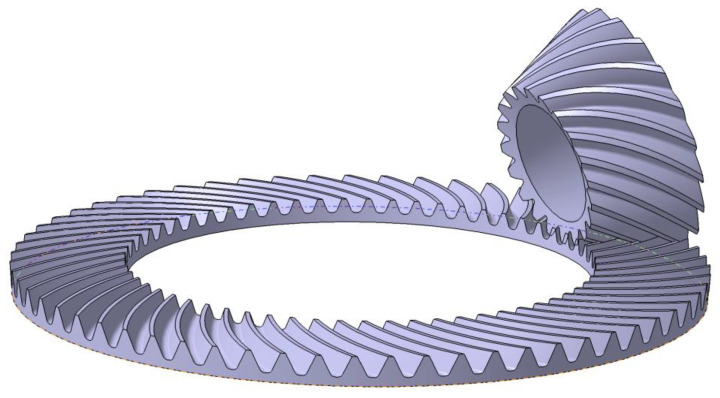
3D model of SBGs.

**Figure 10 materials-14-04848-f010:**
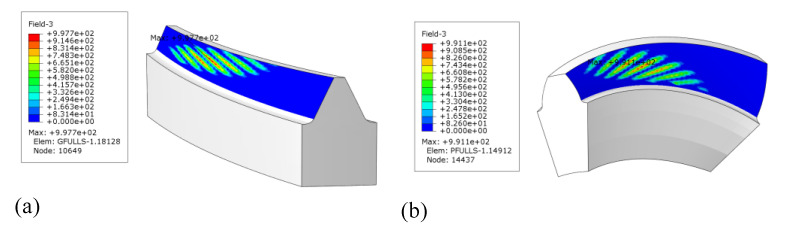
(**a**) Contact area on the gear; (**b**) Contact area on the pinion.

**Figure 11 materials-14-04848-f011:**
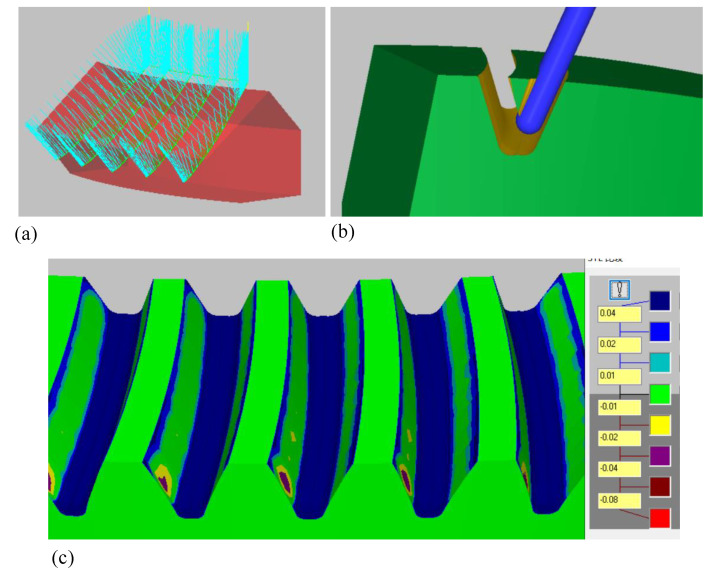
(**a**) Tool path generation by simulation; (**b**) simulation machining of five-axis flank milling; (**c**) tooth surface errors between simulated machining tooth surface and theoretical tooth surface.

**Figure 12 materials-14-04848-f012:**
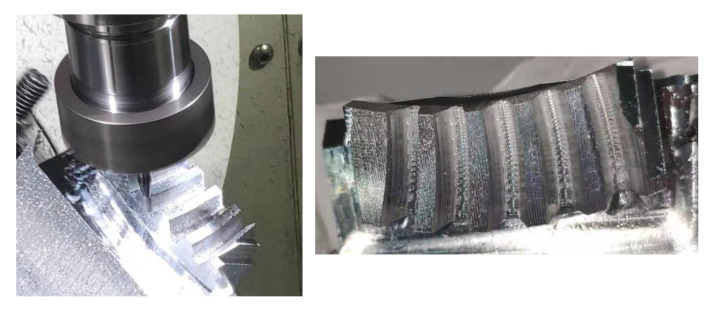
Five-axis flank milling of SBG.

**Figure 13 materials-14-04848-f013:**
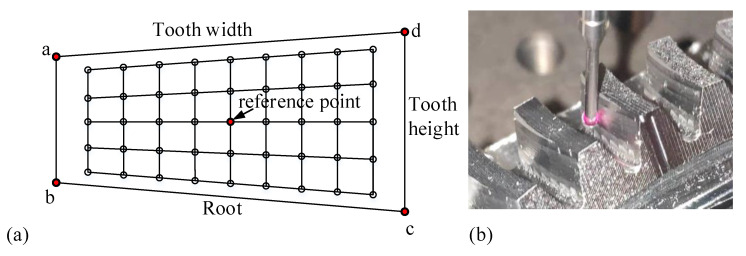
(**a**) Measurement mesh; (**b**) measuring tooth profile errors.

**Figure 14 materials-14-04848-f014:**
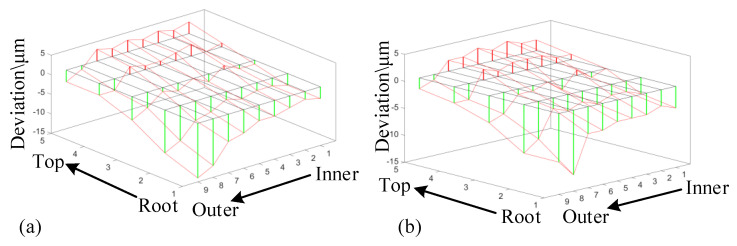
(**a**) Convex side; (**b**) Concave side.

**Table 1 materials-14-04848-t001:** Main data of a face-milled generated SBG.

Blank Data			
Parameter	Value	Parameter	Value
Gear tooth number	61	Pinion tooth number	20
Module	4.8338	Shaft angle	90.0000°
Pinion handle	Right hand	Mean spiral angle	32°
Face width	27.5000 mm	Clearance	1.0300 mm
Outer addendum	1.7600 mm	Outer dedendum	7.6700 mm
Face angle	76.1167°	Root angle	69.5833°
**Blade Data**			
**Parameter**	**Value**	**Parameter**	**Value**
Average radius	63.5000 mm	Point width	2.5400 mm
Pressure angle	22.0000°	Fillet radius	1.5000 mm
**Machine-Settings**			
**Parameter**	**Value**	**Parameter**	**Value**
Radial setting	64.3718 mm	Cradle angle	−56.7800°
Sliding base	0.0000 mm	Machine center to back	0.0000 mm
Blank offset	−0.2071 mm	Machine root angle	69.5900°
Roll ratio	1.0323	Modified roll coefficients	0.0000

## Data Availability

The data presented in this study are available on request the corresponding author.

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
