# Peer review of "An Efficient Approach to the Five-Axis Flank Milling of Non-Ferrous Spiral Bevel Gears"

_materials, 2021, doi:10.3390/ma14174848_

Round 1

Reviewer 1 Report

  1. The Five-axis flank milling technology for bevel gears is already widely used in the industry. The differences between the proposed method and those already used in industry should be shown.
  2. Please describe where aluminum bevel wheels are used?
  3. Line 9-10 Bevel wheels are also produced on 5-axis milling machines for large dimensions above 1.5 meters, for which there are practically no specialized machines.
  4. In terms of the description of the state of the art (Introduction), the following should be done:

- a review of world literature, not only Chinese

- provide the methods of industrial wheel cutting on specialized machines (Gleason and Kligelnberg) and on DMG 5-axis machines, Heller from Gleason (up-gear method).

- mention the correction systems of incised teeth KIMOS and CAGE.

  1. Figure 1 - Toe? Heel? is it the outer or inner diameter of the circle?
  2. Equation 4 - Should we explain what is M?
  3. Do formulas 3-11 take into account the involute contours of the teeth of the bevel gears?
  4. Figure 5 How does the surfaces shown in the drawings relate to the actual surface of the bevel wheels?
  5. Figure 9 Describe which method of incision (Formate, CycloCut, Zyklopalloid, other) corresponds to the presented model
  6. Spacing between words should be corrected in the text, eg Fig 10 "(a) Contact area on the gear;"
  7. Fig. 10 Why is the contact trace in the form of 7 parts? How is it that the trace is not in the middle?
  8. Chapter 5 Why were only the 5 teeth of the wheel cut? Is the pinion cut? It would be correct to cut both wheels and check the trace of cooperation on the tester.
  9. In Fig. 13 the height of the teeth is constant and in the Gleason method the teeth are of variable height. Explain the method used to present the measurement result.
  10. In conclusion, it should be shown that the proposed method is more efficient or comparable to the already used companies such as DMG-Mori or Heller.
  11. Supplement the literature with the item, eg German, American, concerning the discussed technology.
  12. For GR200V A15SH machine, please specify the manufacturer.
  13. In chapter 5, the aluminum grade should be specified.
  14. The discussion of the obtained experimental results should be extended.
  15. Table 1 What were the corrections for bevel gears?

Reviewer 2 Report

In this work, the authors developed an efficient approach to the five-axis flank milling of non-ferrous spiral bevel gears. The research appears to be efficiently done and appropriately reported, however, the standard of English is acceptable only needs few corrections. Nevertheless, there are some questions and corrections that must be answered to improve and complete the document.

The authors didn’t use the journal (Materials from MDPI) template. Please, change your formatting for that template.

Abstract section: The abstract is a little bit confuse and missis some information like more results and conclusions, I suggest to authors follow these rules:

  1. One or two sentences on BACKGROUND
  2. Two or three sentences on METHODS
  3. Less than two sentences on RESULTS
  4. One sentence on CONCLUSIONS

Introduction section: In this section, the authors don’t indicate the novelty of their work. what is the innovation of your work when compared with the other researchers? The "Knowledge gap to be filled"? In this introduction, the authors must describe or indicate the work that will be done to test their "hypothesis".

Lines 41 and 183. when is used an abbreviation for the first time it must be indicated its meaning. Please, write the meaning of “TCA” (line 41) and “CMM”.

Lines 102-103. The authors wrote, “… h calculated by numerical methods.”. Which numerical methods? Please, explain in more detail about these numerical methods.

Lines 164-169. In this paragraph the authors made a brief presentation about their numerical simulation, however, they must add much more information, namely: How many elements did they use in their simulations? Which kind of element was used in the simulation? What boundary conditions were used? How long did the simulations take? Quality of mesh: skewness and orthogonal quality? What converge criteria were used?

Legend of Figure 11 c) “Tooth surface … “. How did they determine the errors? What software did they use to obtain these errors?

Line 188. Please change “20um” to “20 µm”.

Line 190. The authors wrote, “The error fluctuation value is in a reasonable range, which …”. What is a "reasonable range" for the authors?

Round 2

Reviewer 1 Report

I have no further objections to the work.

Reviewer 2 Report

The second version of the manuscript improved significantly when compared with the first version. So, in my opinion, the manuscript can be accepted for publication.

This manuscript is a resubmission of an earlier submission. The following is a list of the peer review reports and author responses from that submission.